# Linear growth and mid-childhood cognitive outcomes in three birth cohorts of term-born children: an approach to integrating three growth models to explore critical windows

Michael Leung [ID],[1,2] Aditi Krishna,[2,3] Seungmi Yang [ID],[4] Diego G Bassani,[2,5,6] Daniel E Roth[2,5,6]

For numbered affiliations see end of article.

**Correspondence to**
Michael Leung;
michael_leung@g.harvard.edu

## ABSTRACT

**Objective** To illustrate that a mediation framework can help integrate inferences from three growth models to enable a comprehensive view of the associations between growth during specific developmental windows and mid-childhood IQ.

**Design** We analysed direct and indirect associations between mid-childhood IQ and length/height growth in five early-life age intervals bounded by conception, birth, early, mid and late infancy, and mid-childhood using estimates from three growth models (lifecourse, conditional change and change score) applied to three historical birth cohorts.

**Participants and setting** 12 088 term-born children from the Collaborative Perinatal Project (CPP) in the USA (n=2170), the Promotion of Breastfeeding Intervention Trial (PROBIT) in Belarus (n=8275) and the Cebu Longitudinal Health and Nutrition Survey (CLHNS) in the Philippines (n=1643).

**Primary outcome measure** Mid-childhood IQ.

**Results** Our analyses revealed cross-cohort and cross-interval variations in the direct and indirect effects of foetal and early childhood physical growth on mid-childhood IQ. For example, in CPP, there was a direct association of prenatal growth with IQ that was not evident in the other cohorts, whereas in PROBIT and CLHNS, we observed that foetal and early growth-IQ associations were mediated through size in later periods.

**Conclusion** Lifecourse, conditional change and change score growth models yield complementary inferences when appropriately interpreted. Future longitudinal studies of associations of early-life growth with later outcomes would benefit from adopting a causal mediation framework to integrate inferences from multiple complementary growth models.

## INTRODUCTION

Three *Lancet* series on child development[1–3] have identified constrained physical growth in early-life, from conception to age 2 years, as one of the key risk factors for impaired cognitive achievement among children from low-income and middle-income countries.

### Strengths and limitations of this study

► The use of three large prospective birth cohorts of term-born children with serial anthropometry data enabled the exploration of critical windows using a comparative, multimodel approach.

► Our analysis was restricted to term-born children to enable the application of the WHO Child Growth Standards, so our results are only generalisable to children born at term.

► There was likely differential error with respect to anthropometric measures across cohorts and age groups that may have affected the decomposition of the total effect of growth on subsequent cognitive achievement.

However, many population-based studies of the association between growth and cognition have relied on single or repeated cross-sectional measures of body length/height rather than longitudinal assessment of growth trajectories, providing limited insights into particularly sensitive growth phases that may be strongly associated with childhood cognitive scores.

Estimates of the association between growth and later outcomes in longitudinal studies commonly use one of three statistical approaches, which we refer to here as the lifecourse model, conditional change model or the change score model[4–7] (table 1). The relative advantages of each approach have been previously described,[4 8] yet it is now recognised that the three models are algebraically interrelated and yield contrasts that are complementary rather than contradictory.[7 9]

Causal mediation analysis helps provide a coherent framework for understanding the complementarity of these models.[9] In the

**Table 1** Summary of three commonly used growth models

| Growth model | Formulation | Interpretation of model coefficients |
|---|---|---|
| Lifecourse | $E\left[Y\vert LAZ_j, C_k\right] = \alpha_0 + \alpha_j^T LAZ_j + \alpha_k^T C_k$ <br> Outcome $Y$ is regressed on LAZ measured at age $j$ ($LAZ_j$), and $k$ potential confounders ($C_k$). | ▶ $\alpha_0$ is the expected value of $Y$ when $LAZ_j$ and $C_k$ are all equal to 0. <br> ▶ $\alpha_j$ is the direct effect of growth in the interval preceding age $j$. <br> ▶ $\alpha_k$ are nuisance parameters associated with the conditional density of $Y$ given $LAZ_j$. |
| Conditional change | $E\left[Y\vert LAZ_0, cLAZ_j, C_k\right] = \beta_0 + \beta_1 LAZ_0 + \beta_j^T c\Delta LAZ_j + \beta_k^T C_k$ <br> Outcome $Y$ is regressed on LAZ at birth ($LAZ_0$), conditional measures of growth between the start and end of each interval ($c\Delta LAZ_j$) and $k$ potential confounders ($C_k$). Conditional growth is the difference between the observed and expected LAZ at the end of the interval, where the expected value is based on size at the beginning of the interval, and can be derived by calculating $\varepsilon_{ij}$, the residual for child $i$ at age $j$, after regressing size at the end of the interval on size at the beginning of the interval. | ▶ $\beta_0$ is the expected value of $Y$ when, $c\Delta LAZ_j$ and $C_k$ are all equal to 0. <br> ▶ $\beta_1$ is the interval-specific total effect of prenatal growth (ie, sum of the direct effect of prenatal growth and the indirect effect via later size). <br> ▶ $\beta_j$ is the interval-specific total effect of growth in the interval preceding age $j$ (ie, sum of the direct effect of interval-specific growth and the indirect effect via later size). <br> ▶ $\beta_k$ are nuisance parameters associated with the conditional density of $Y$ given $LAZ_0$ and $c\Delta LAZ_j$. |
| Change score | $E\left[Y\vert LAZ_0, LAZ_j, C_k\right] = \theta_0 + \theta_1 LAZ_0 + \theta_j^T \Delta LAZ_j + \theta_k^T C_k$ <br> Outcome $Y$ is regressed on LAZ at birth ($LAZ_0$), the change score (absolute change in size between the start and end of each interval ($\Delta LAZ_j$)), and $k$ potential confounders ($C_k$). | ▶ $\theta_0$ is the expected value of $Y$ when $LAZ_0$ $\Delta LAZ_j$, and $C_k$ are all equal to 0. <br> ▶ $\theta_1$ is the cumulative direct effect of growth in the prenatal and all subsequent intervals included in the model. <br> ▶ $\theta_j$ is the cumulative direct effect of growth in the interval preceding age $j$ and all subsequent intervals included in the model. <br> ▶ $\theta_k$ are nuisance parameters associated with the conditional density of $Y$ given and $\Delta LAZ_j$. |

LAZ, length-for-age z-scores.

lifecourse model where all of length/height measured at different ages are modelled simultaneously, the coefficient for a given age represented by $t_j$ can be interpreted as the *direct* effect of growth in the preceding interval, or the interval between $t_{j-1}$ and $t_j$. Coefficients from the conditional change models are estimates of the *total* effect of growth in each interval, which combines the *direct* effect of growth in each interval and the *indirect* effect of the resultant attained size in subsequent intervals. Because the measure of growth in each interval is conditioned on attained size at the end of the preceding interval, each coefficient of growth is independent of the other growth coefficients in the model.[4 5] The change score model is similar to the conditional change model, but the outcome is regressed on the interval-specific starting size and the *absolute* differences between size at the start and end of each subsequent interval; consequently, growth in one interval may be correlated with growth in other intervals. In the change score model, each coefficient represents the cumulative direct effect of growth in the current interval and in all subsequent intervals included in the model; for example, the coefficient for the initial size term (eg,

birth length) reflects the cumulative direct effects of foetal and postnatal growth across all intervals.

It is important to acknowledge that although we refer to these estimates as *effects*, where we borrow language from causal mediation, they should be interpreted as associations. It is unlikely that the relationship between linear growth and brain development is causal as we are not aware of any convincing biological mechanism for this effect. Instead, it is more likely that linear growth and brain development share the same underlying determinants, such as maternal and infant undernutrition, which is common in these settings.[1–3] Through this lens, length/height can be viewed as a proxy (although imperfect) for exposure to undernutrition, where an association between early-life linear growth and cognitive scores would be indicative of a nutritional intervention that prevents both slow growth and faltering in brain development (ie, it would represent the effect of intervening nutritionally, as opposed to the effect of physical growth itself).[10]

Here, we demonstrate the integration of estimates from all three growth models in a mediation analysis framework to provide a comprehensive view of the

associations between growth in early life and cognitive achievement at ages 5–9 years among term-born children.

## METHODS
### Data sources
This study uses data from three historical birth cohorts that include anthropometric data from birth to mid-childhood (ages 5–9 years) and cognitive assessments at ages 5–9 years, namely; the Collaborative Perinatal Project (CPP) in the USA (1959–1964),[11] the Promotion of Breastfeeding Intervention Trial (PROBIT) in Belarus (1996–1997)[12] and the Cebu Longitudinal Health and Nutrition Survey (CLHNS) in the Philippines (1983–1984).[13] All data were made available by the Healthy Birth, Growth and Development Knowledge Integration (HBGDki) initiative funded by the Bill and Melinda Gates Foundation.[14]

### Linear growth exposures
Foetal and infant growth variables were derived as required for the lifecourse, conditional change and change score models, as described in table 1 and in detail elsewhere.[4 7] Briefly, we derived length-for-age z-scores (LAZ) using the WHO Child Growth Standards (WHO-GS).[15] All models use LAZ at birth ($LAZ_0$) as a relative measure of foetal growth from conception to birth. Postnatal growth during the first year was partitioned into early, mid and late infancy periods, with different cut-offs depending on the timing of measurement in each cohort. In CPP and CLHNS, these periods corresponded to 0–4 months ($LAZ_1$), 4–8 months ($LAZ_2$) and 8–12 months ($LAZ_3$), while in PROBIT, the analogous periods spanned 0–3 months ($LAZ_1$), 3–9 months ($LAZ_2$) and 9–12 months ($LAZ_3$). Finally, in all three cohorts, we used LAZ at ages 6.5 years (PROBIT study), 7 years (in the CPP cohort) and 8.5 years (in the CLHNS cohort) to represent the period of postnatal growth from 1 year to mid-childhood ($LAZ_4$).

### Cognitive achievement in mid-childhood
Mid-childhood cognitive achievement was assessed using within-cohort IQ z-scores (IQz). The mean IQ of the sample was subtracted from each individual IQ score and the result was divided by the SD of the IQ distribution of the cohort to produce internally standardised scores. IQ was assessed in mid-childhood using the Wechsler Intelligence Scale for Children in CPP,[16] the Wechsler Abbreviated Scale of Intelligence in PROBIT[12] and the Philippines Non-Verbal Intelligence Test in CLHNS.[13] Global measures of IQ were used—full-scale IQs from CPP and PROBIT, and non-verbal IQ score from CLHNS—and all were age-standardised prior to generating the IQz.

### Child, parental and household characteristics
Covariates included child, parental and household characteristics measured at birth. Due to differences in data availability, there is slight variation in the specific sets of covariates used in the models across the three cohorts. Child sex, maternal and paternal age (years), maternal height (cm) and maternal and paternal education (years of completed schooling in CPP and CLHNS; credential-based in PROBIT) were common covariates across the cohorts. Estimates using the CPP dataset also included the following covariates: gestational age categorised as early term (37 to <39 weeks), full term (39 to <41 weeks) and post-term (≥41 weeks); and parity (nulliparous, primiparous and multiparous). The analysis of the PROBIT cohort includes gestational age (categorised as above); number of older siblings (none, one, two or more); paternal height (cm); maternal weight (kg); maternal body mass index ($kg/m^2$); maternal occupation (non-manual, manual, unemployed) and paternal occupation (non-manual, manual, unemployed, unknown). Estimates using the CLHNS dataset includes adjustment for parity (categorised as above), and within-cohort quintiles of socioeconomic status derived from an asset-based index.[13]

### Statistical analyses
We examined associations between interval-specific linear growth in early life and mid-childhood IQz for each cohort using all three growth models, adjusted for baseline values of the child, parental and household covariates listed above. Visual inspections revealed that the relationship between LAZ and IQz was approximately linear throughout the full range of LAZ (ie, without an apparent inflection point), as has been previously reported[17]; therefore, all analyses included LAZ as a continuous variable.

Lifecourse, conditional change and change score models were used to estimate the interval-specific direct effects, interval-specific total effects and cumulative direct effects, respectively, with corresponding 95% CIs. We also assessed statistical heterogeneity of the lifecourse and conditional change coefficients across the three cohorts, and across the five growth intervals using the $I^2$ statistic.[18]

For consistent mediation models (ie, models in which the total effect and the direct effect have the same sign), we used a difference-in-coefficients approach to mediation analysis[19 20] to estimate the proportion of the total growth-cognition association for a given interval that was mediated by size in later intervals. The proportion was estimated by dividing the indirect effect (difference between the total effect and the direct effect) by the total effect for each period within each cohort.

Children were excluded from all analyses if they: (1) lacked complete anthropometry data for a scheduled assessment, (2) did not have cognitive achievement assessed in mid-childhood, (3) were born preterm (to enable comparisons of growth modelled using the WHO-GS which should not be directly applied to children born preterm in epidemiological research[21]), (4) had extreme LAZ based on the WHO-GS (LAZ>6 or LAZ<−6) or (5) had incomplete data on any of the baseline covariates.

Analyses were performed using STATA V.13 (Stata Corporation). Analyses of these datasets from the HBGDki repository was approved by the Hospital for Sick Children (Toronto, Canada).

### Patient and public involvement

Patients or the public were not involved in the design of the study.

## RESULTS

A total of 12 088 children met the inclusion criteria and were included in the analyses: 2170 from CPP, 8275 from PROBIT and 1643 from CLHNS (online supplementary figure 1). Availability of data related to child, parental and household characteristics differed across the cohorts (table 2). Mean LAZ trajectories from birth to mid-childhood, which have been previously described,[22–25] differed across the three cohorts; in particular, children in CLHNS experienced substantial postnatal growth faltering compared with CPP and PROBIT (figure 1; online supplementary table 1).

Estimates of the associations between linear growth in discrete intervals and mid-childhood IQz were derived from the three multivariable-adjusted growth models (table 3). In all three cohorts, the cumulative direct effects of foetal and postnatal growth across all intervals were consistently positive but of modest magnitude, not exceeding 0.2 IQz per unit increase in LAZ. In each interval, total effects were consistently positive but small. However, the patterns of direct effects differed across the three cohorts: there was a positive direct effect of foetal growth (conception to birth) on IQz in CPP but not in PROBIT or CLHNS. In the three infancy intervals, starting with the interval between birth and early infancy, the direct effects were inconsistent in both CPP and PROBIT, and null in all intervals for CLHNS. From late infancy to mid-childhood, there was no direct effect of growth on IQz in CPP but positive direct effects in PROBIT and CLHNS. Differences across cohorts and intervals in the magnitude of the effects were supported by heterogeneity statistics derived from meta-analyses (online supplementary table 2).

Estimates of the proportions of the effects that were mediated by size in later intervals are also presented in table 3. In CPP, the proportion of the total effect of growth in each interval that was mediated by being longer/taller in later intervals was modest, but highest in the prenatal (39%) and birth-to-early infancy periods (38%). In PROBIT, almost all of the total foetal growth effect (96%) was mediated by larger size in the postnatal period. In CLHNS, the proportion of the total effect of growth in each interval that was mediated through longer/taller size in later intervals was high compared with the other two cohorts (59%–90%).

## DISCUSSION

Most observational epidemiological studies of the effects of growth on later-life outcomes have only used one statistical approach—commonly, researchers have selected the lifecourse, conditional change or change score growth model.[23 26–33] Post-hoc integration of the evidence from such studies is complicated, as the coefficients of the lifecourse, conditional change and change score model represent interval-specific direct effects, interval-specific total effects and cumulative direct effects, respectively; that is, each model yields a contrast that is not directly comparable to the contrasts derived from the other two models.

Here, we demonstrated the combined use of multiple models to generate coherent inferences on the associations between interval-specific physical growth and later cognitive outcomes. In a previous study of a cohort of children from Ethiopia, India, Peru and Vietnam, Georgiadis et al [9] used a similar approach to conclude not only that growth from conception to 8 years (considered in three intervals: conception to 1 year, 1–5 years and 5–8 years) was associated with higher cognitive achievement in mid-childhood, but also that the majority of the estimated association between early growth and mid-childhood cognition was mediated by being taller in subsequent periods. In the present study, we demonstrated an extension of this approach to discrete growth intervals within infancy (birth to 1 year), and in three heterogenous historical birth cohorts. Similar to the findings of Georgiadis et al,[9] we showed that the total effects were consistently positive in all three cohorts, thereby reaffirming the association between early physical growth and later cognitive achievement.[1 2 34–36] In two of the three cohorts (PROBIT and CLHNS), growth in later periods of early childhood was more strongly associated with IQ than growth in infancy or in utero. In both cohorts, we found no direct effect of growth from conception to birth; rather, associations of birth with cognition were mostly mediated by later size, especially in the interval from 1 year to mid-childhood. However, in CPP, foetal growth was more strongly associated with IQ than postnatal growth; we speculate that because postnatal growth was relatively unconstrained (vs prenatal growth) in the US cohort compared with the cohorts in the Philippines and Belarus, foetal growth appeared to be a relatively more important contributor to later cognitive outcomes in the US context.

A key motivation of this study was to demonstrate how epidemiologists can integrate inferences from three commonly used growth models. In future applications, we would advocate for the simultaneous use of two of the models (lifecourse and conditional change models), as estimates of the direct and total effect for a given age interval allow inferences about the proportion of the effects that are mediated by size in later periods. Investigators who elect to use the change score model alone should be careful to correctly interpret its coefficients as estimates of the cumulative direct effect of growth for a given period and all successive periods included in the model. This is a non-intuitive interpretation, and it is important to recognise that the model coefficients

**Table 2** Child, parental and household characteristics of the three birth cohorts

| Characteristics | CPP (n=2170) | PROBIT (n=8275) | CLHNS (n=1643) |
|---|---|---|---|
| **Child** | | | |
| Female, % | 49 | 49 | 47 |
| Term status (weeks) | | | |
| Early term (37 to <39), % | 8.9 | 19 | – |
| Full term (39 to <41), % | 36 | 80 | – |
| Post-term (≥41), % | 55 | 1.2 | – |
| **Parental and household characteristics at birth** | | | |
| Socioeconomic status quintile | | | |
| Low, % | – | – | 19 |
| Lower middle, % | – | – | 18 |
| Middle, % | – | – | 35 |
| Upper middle, % | – | – | 8.5 |
| Upper, % | – | – | 20 |
| Maternal age (years), mean (SD) | 26.8 (5.0) | 24.5 (4.9) | 26.5 (5.9) |
| Maternal race | | | |
| White, % | 79 | – | – |
| Black, % | 20 | – | – |
| Other, % | 0.4 | – | – |
| Maternal height (cm), mean (SD) | 162.9 (6.5) | 164.4 (5.7) | 150.6 (4.9) |
| Maternal weight (kg), mean (SD) | – | 66.4 (12.6) | – |
| Maternal BMI (kg/m$^2$), mean (SD) | – | 24.5 (4.4) | – |
| Maternal education (years), mean (SD) | 13.2 (1.7) | – | 7.0 (3.3) |
| Maternal education | | | |
| University, % | – | 14 | – |
| Partial university, % | – | 52 | – |
| Secondary, % | – | 31 | – |
| <Secondary, % | – | 2.9 | – |
| Maternal occupation | | | |
| Non-manual, % | – | 45 | – |
| Manual, % | – | 34 | – |
| Unemployed, % | – | 20 | – |
| Parity | | | |
| Nulliparous, % | 49 | – | 19 |
| Primiparous, % | 25 | – | 23 |
| Multiparous, % | 26 | – | 59 |
| Number of older siblings | | | |
| None, % | – | 55 | – |
| One, % | – | 37 | – |
| Two or more, % | – | 8.8 | – |
| Paternal age (year), mean (SD) | 29.6 (5.9) | 27.5 (5.1) | 29.1 (6.7) |
| Paternal height (cm), mean (SD) | – | 175.9 (6.6) | – |
| Paternal education (years), mean (SD) | 14.0 (2.2) | – | 7.2 (3.4) |
| Paternal education | | | |
| University, % | – | 14 | – |

| Characteristics | CPP (n=2170) | PROBIT (n=8275) | CLHNS (n=1643) |
|---|---|---|---|
| Partial university, % | – | 48 | – |
| Secondary, % | – | 37 | – |
| <Secondary, % | – | 2.0 | – |
| Paternal occupation | | | |
| Non-manual, % | – | 30 | – |
| Manual, % | – | 56 | – |
| Unemployed, % | – | 14 | – |
| Unknown, % | – | 0.5 | – |

**Table 2** Continued

BMI, body mass index; CLHNS, Cebu Longitudinal Health and Nutrition Survey; CPP, Collaborative Perinatal Project; PROBIT, Promotion of Breastfeeding Intervention Trial.

cannot be interpreted as estimates of the discrete effects of growth in the interval from which they were derived. As demonstrated here, the change score model coefficients for birth size represented the theoretical maximum direct effect of a one-unit increase in LAZ on IQz in all three cohorts; concordant with expectations, the coefficients then decrease in magnitude as the period of growth approaches the time of outcome assessment. Investigators may be tempted to attribute the strongest effects of growth to the earliest ages; however, coefficients from the other two models revealed that the direct and total effects are largest from mid-infancy to late infancy in CPP, and late infancy to mid-childhood in both PROBIT and CLHNS.

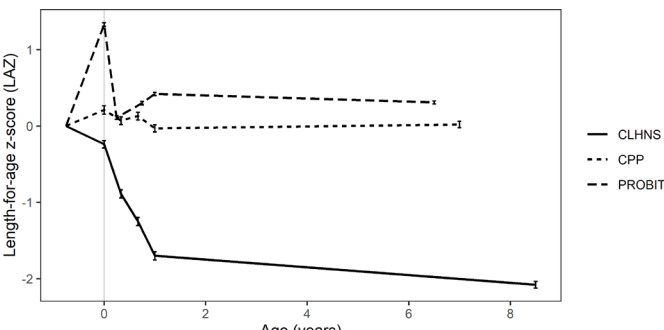

**Figure 1** Mean length-for-age z-scores (LAZ) from birth to mid-childhood in the Collaborative Perinatal Project (CPP) in the USA (n=2170), the Promotion of Breastfeeding Intervention Trial (PROBIT) in Belarus (n=8275) and the Cebu Longitudinal Health and Nutrition Survey (CLHNS) in the Philippines (n=1643). For each cohort, estimates and 95% CIs (vertical error bars) are shown at the end of age intervals of interest: birth, early infancy (3 or 4 months), mid-infancy (8 or 9 months), late infancy (12 months) and mid-childhood. Estimates for each cohort are connected by straight lines to represent population trajectories, assuming linearity in all intervals for visual simplicity. Prenatal LAZ trajectories are theoretical, assuming that foetal growth starts at a mean LAZ of 0 in all populations and gestational age distributions are identical across cohorts.

If a research study aims to estimate the association of physical growth in only one age interval with a later outcome, rather than consider multiple intervals to identify a developmental window in which growth is most strongly associated, then there is no need to use multiple models, as all three models will yield the same contrast. In our analyses, this was evident when evaluating the last growth interval in each cohort (ie, the interval closest in time to the assessment of cognitive scores) as the direct, total and cumulative effects are identical in all three models.

A strength of our study was the use of three large prospective birth cohorts of term-born children with serial anthropometry data (including multiple measurements in infancy). It enabled the exploration of critical windows using a comparative, multimodel approach and demonstrated the context-dependency of conclusions. However, there are several limitations that should also be acknowledged. First, we assumed no information bias, yet the presence of differential errors with respect to anthropometric measures both across cohorts and age groups was likely and may have affected our within-cohort and between-cohort comparisons of the decomposition of the effect of growth on subsequent cognitive achievement. Second, even though our eligibility criteria of being born at term was necessary to enable the application of the WHO-GS, the resulting effect estimates are only generalisable to term-born children. Finally, we could not partition growth from 1 year to mid-childhood into smaller intervals due to anthropometric assessment schedules used in each study. Most notably, we could not isolate growth during the second postnatal year of life, which may have yielded further insights.[3]

Although we refer to the *effects* of growth in this paper, it is important to highlight that we are using LAZ as a proxy for underlying nutritional status. For any of the growth-cognition associations we describe above to represent the effect of intervening on nutritional status, we assume that the following conditions for causal mediation analysis hold: (1)

**Table 3** Multivariable adjusted estimates and 95% CIs of the associations between early childhood growth and mid-childhood cohort-specific IQ z-score in three birth cohorts

| Growth interval | Lifecourse model (direct effect) | Conditional change model (total effect) | Change score model (cumulative direct effect) | Proportion mediated* |
|---|---|---|---|---|
| CPP† (n=2170) | | | | |
| Conception to birth | 0.05 (0.01 to 0.08) | 0.08 (0.05 to 0.11) | 0.16 (0.12 to 0.20) | 0.39 |
| Birth to early infancy | 0.04 (−0.01 to 0.08) | 0.06 (0.02 to 0.10) | 0.11 (0.06 to 0.16) | 0.38 |
| Early infancy to mid-infancy | −0.03 (−0.08 to 0.02) | 0.03 (−0.02 to 0.07) | 0.08 (0.02 to 0.13) | – |
| Mid-infancy to late infancy | 0.07 (0.02 to 0.12) | 0.09 (0.04 to 0.14) | 0.11 (0.06 to 0.16) | 0.24 |
| Late infancy to mid-childhood | 0.04 (−0.01 to 0.09) | 0.04 (−0.01 to 0.09) | 0.04 (−0.01 to 0.09) | 0.0‡ |
| PROBIT§ (n=8275) | | | | |
| Conception to birth | 0.001 (−0.10 to 0.10) | 0.03 (−0.06 to 0.12) | 0.15 (0.07 to 0.23) | 0.96 |
| Birth to early infancy | 0.05 (−0.02 to 0.13) | 0.06 (−0.01 to 0.12) | 0.15 (0.08 to 0.22) | 0.02 |
| Early infancy to mid-infancy | −0.04 (−0.10 to 0.01) | 0.003 (−0.04 to 0.04) | 0.09 (0.04 to 0.15) | – |
| Mid-infancy to late infancy | 0.02 (−0.04 to 0.07) | 0.07 (0.01 to 0.13) | 0.14 (0.07 to 0.21) | 0.73 |
| Late infancy to mid-childhood | 0.12 (0.07 to 0.18) | 0.12 (0.07 to 0.18) | 0.12 (0.07 to 0.18) | 0.0‡ |
| CLHNS¶ (n=1643) | | | | |
| Conception to birth | 0.01 (−0.05 to 0.06) | 0.06 (0.02 to 0.11) | 0.20 (0.14 to 0.26) | 0.89 |
| Birth to early infancy | 0.04 (−0.02 to 0.10) | 0.10 (0.05 to 0.15) | 0.19 (0.13 to 0.25) | 0.59 |
| Early infancy to mid-infancy | 0.02 (−0.06 to 0.10) | 0.08 (0.02 to 0.14) | 0.15 (0.08 to 0.22) | 0.77 |
| Mid-infancy to late infancy | 0.01 (−0.07 to 0.08) | 0.08 (0.00 to 0.15) | 0.13 (0.04 to 0.22) | 0.90 |
| Late infancy to mid-childhood | 0.12 (0.06 to 0.18) | 0.12 (0.06 to 0.18) | 0.12 (0.06 to 0.18) | 0.0‡ |

*For consistent mediation models, proportion mediated was calculated by dividing the indirect effect by the total effect for a given growth interval (ie, proportion of the total effect mediated by subsequent size).

†Adjusted for child sex, maternal height, maternal and paternal education, gestational age category, maternal and paternal ages, and parity.

‡Indirect effect is 0 as growth in the last interval is not mediated by later size.

§Adjusted for child sex, maternal height, maternal and paternal education, gestational age category, paternal height, maternal weight, maternal body mass index, number of older siblings and maternal and paternal occupation.

¶Adjusted for child sex, maternal height, maternal and paternal education, maternal and paternal ages, parity, and socioeconomic status.

no unmeasured time-invariant confounders, (2) no unmeasured time-varying confounders and (3) no effect measure modification by any of the mediating variables in the causal structure. For the first assumption, confounding (residual or unmeasured) can never be ruled out. Although we controlled for an extensive, although different across the three cohorts, panel of baseline covariates (ie, child, parental and household attributes), we may not have controlled for all relevant confounders. For example, we did not control for maternal smoking,[37–40] which was highly prevalent in the USA in the 1950s–1960s.[41] However, maternal smoking is only associated with mid-childhood cognitive achievement through its association with socioeconomic status, as shown by previous research in the same cohorts examined in the present study,[42 43] and in other settings.[44 45] One possible explanation of this finding is that maternal smoking affects early-life brain development,[39 40] but the effects diminish through childhood, such that by mid-childhood (ie, timing of outcome assessment) smoking is just a behavioural trait embedded in a

broader constellation of social factors that influence child brain development, such as early stimulation and learning opportunities.[3] Thus, our control for parental education, employment and socioeconomic status quintile in our models would be sufficient to block this backdoor path (borrowing language from causal graphs[46]) through maternal smoking (and any factors associated with mid-childhood cognitive achievement that are largely shaped by a mother's social conditions). For the second and third conditions, none of the models as they are commonly formulated can address either time-varying confounding or potential exposure-mediator interactions, but potential violations of these conditions could be considered through sensitivity analyses (eg, one could try assess the magnitude of confounding and/or effect measure modification that would be needed to explain away the effect estimates).

We recognise that this approach has its limitations, and that there are many alternative methods to model growth[6]; for example, a distributed lag model may provide insights into sensitive windows by modelling

direct effects as a smooth function of age.[47] However, the goal of this paper was to show that estimates reported from the lifecourse, conditional change and change score models, which are among the three most commonly used growth models by epidemiologists, nutritional scientists and developmental origins of health and disease researchers,[6] can be synchronised post-hoc using a mediation framework to aid in the interpretation of studies investigating critical growth windows. With this approach, we showed that linear growth is weakly but consistently associated with subsequent cognitive achievement across diverse settings but did not find evidence to support the identification of a universal 'sensitive' window, as the magnitude of the growth-cognition association during specific developmental windows varied by setting. Cognitive achievement in mid-childhood was most strongly associated with foetal and early growth in the USA in 1959–1971, and with growth in later age intervals in Belarus in 1996–2004 and the Philippines in 1983–1993. Although we used growth in length/height and IQ for our illustrative example, this approach can be extended to investigating critical windows of growth using other anthropometric measures, such as weight or head circumference (which may be more relevant for cognitive achievement[48 49]) and other later-life health outcomes.

**Author affiliations**
¹Epidemiology, Harvard University TH Chan School of Public Health, Boston, Massachusetts, USA
²Centre for Global Child Health, The Hospital for Sick Children, Toronto, Ontario, Canada
³Iris Group, Chapel Hill, North Carolina, USA
⁴Epidemiology, Biostatistics and Occupational Health, McGill University Health Centre, Montreal, Quebec, Canada
⁵Department of Paediatrics, The Hospital for Sick Children, Toronto, Ontario, Canada
⁶Dalla Lana School of Public Health, Toronto, Ontario, Canada

**Acknowledgements** We thank our colleagues in the Healthy Birth, Growth and Development knowledge integration (HBGDki) consortium, and HBGDki staff at the Bill and Melinda Gates Foundation. We also greatly appreciate the work of the original cohort investigators and participants who enabled the generation of these datasets.

**Contributors** ML, AK, SY, DGB and DER designed the study; ML, AK and SY analysed data and performed statistical analysis; ML, AK, SY, DGB and DER wrote the paper; ML and DER have primary responsibility for final content. All authors read and approved the manuscript.

**Funding** This work was supported by the Bill and Melinda Gates Foundation (OPP1133178).

**Competing interests** None declared.

**Patient consent for publication** Not required.

**Provenance and peer review** Not commissioned; externally peer reviewed.

**Data availability statement** Data may be obtained from a third party and are not publicly available. All data are available at the Healthy Birth, Growth and Development Knowledge Integration (HBGDki) initiative funded by the Bill and Melinda Gates Foundation.

**ORCID iDs**
Michael Leung http://orcid.org/0000-0003-4831-9566
Seungmi Yang http://orcid.org/0000-0002-1189-6821

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
