## [Reviewer comments · BMJ Open]

ARTICLE DETAILS

TITLE (PROVISIONAL)	Linear growth and mid-childhood cognitive outcomes in three birth cohorts of term-born children: an approach to integrating three growth models to explore critical windows
AUTHORS	Leung, Michael; Krishna, Aditi; Yang, Seungmi; Bassani, D; Roth, Daniel

VERSION 1 – REVIEW

REVIEWER	Thomas ZiesemerH.W. Maastricht University, Netherlands
REVIEW RETURNED	21-Feb-2020

GENERAL COMMENTS	Linear growth and mid-childhood cognitive outcomes in three birth cohorts of term-born children: an approach to integrating three growth models to explore critical windows Conceptually and from the perspective of control variables and policy, it is surprise that the emphasis is completely on length/height whereas weight should do the same. In addition, it is easier to have a policy impact on weight rather than on height. Including it in the set of variables could reveal which of the variables is explaining IQs better. At least, it should be explained why weight is ignored altogether. p.9: If the relation is only approximately linear, it could be useful to learn how results change if one or both of the variables had the form of natural logarithms. If I understand this correctly, IQs are measured only once in each of the samples, and therefore this is a cross section analysis. The text about longitudinal analysis, in contrast seems to indicate that there is a time dimension, leading to a panel analysis. This could be pointed out more clearly. Stata can provide VIF (variance inflation factors) which signal collinearity with control variables. In case of collinearity, the coefficients may change and this may disturb the view on 'sensitive windows'. The standard recipe than is to drop the weaker variables. In connection with the issue of weight, indicated above, this could provide much more clarity. If close to windows, the use of polynomial distributed lags may bring the shape forward. If this would be similar across the three samples, it would really be an interesting result. Most importantly, Table 3 should have or t (z) values, and indicator to let us see something of the goodness of fit. In sum, there is much room of improvement here. In economics, it would be standard to use this before publication. In connection with differences in protocol procedure the editor should decide what the authors have to do and what is a suggestion for future research. Minor issues
---

	Define 'stature' It remains unclear why the initial value is interpreted as revealing information on postnatal growth. That comes after the initial values and would have to be excluded as an extra variable as it is done in the second and third approach. If I mis-understand this, it would be helpful to get to know what medical sciences think about the causal role of interval 1-4 for LAZ.
--	---

REVIEWER	May Ching Soh King's College London, United Kingdom University of Auckland, New Zealand.
REVIEW RETURNED	26-Feb-2020

GENERAL COMMENTS	I enjoyed reading your paper and found it exceedingly well written with a balanced view of the subject. You have adequately discussed the possible drawbacks of the study. I have very little to add to this. However, a specialist review for the statistical analysis of the paper would be advised.
---

REVIEWER	Dieter Wolke University of Warwick United Kingdom
REVIEW RETURNED	29-Apr-2020

GENERAL COMMENTS	This study sets out to compare the relationship of the method of computing growth using three different models and the relationship to IQ in mid childhood. In a nutshell, all three models: life course, conditional change and change score came in two cohorts to more or less similar findings; i.e. they mostly indicated in the three cohorts studied similarly the time periods when growth had the most effect. Quite different findings were found between the Lifecourse model and the other two models in their association to IQ. In the CLHNS the findings were quite different. One could argue that it would not matter what data are put in as long as the three models use the same data. But the authors also compare findings across the three cohorts that were sampled across different areas. And the critical windows found differ for these three cohorts. The question is how can that be explained?  1. To have a fair comparison, the three cohorts should be harmonised. Although differently measured this was done via z-scores for growth (LAZ) and IQ outcome via WHO standards or within cohort, respectively. However, different confounders were included and those which measured a similar construct were not harmonised to measure the same. If they cannot be harmonised they should be discarded to not compare "apples with bananas". 2. There are major confounders that may explain differences – such as the effect of prenatal growth in the oldest sample (USA). A major confounder related to growth is maternal smoking or passive smoking exposure. The rate of smoking was the highest in the USA in the 1950s and 60s in documented history (https://www.cdc.gov/mmwr/preview/mmwrhtml/mm4843a2.htm). Thus foetuses may have been strongly compromised. Thus smoking in all three cohorts as well as breastfeeding rates (as related to growth) should have been included as confounders or critically discussed why they were not included. 3. The number of those lost to follow-up is large in all samples but this is not so concerning as this study is about comparing different predictive models. And previous research has found relatively unbiased prediction despite large loss in follow-up. Nevertheless, in
---

	STATA missing data for some of the variables could have been computed with the MICE function. 4. The most concerning is there should be some explanation why LAZ should be related to IQ and via what mechanism. IQ is a measure of brain function and how should weight and height relate to it – which biological or social mechanisms? Intelligence happens in the brain and thus, of the different anthropometric measures, head circumference is the closest measure related to brain growth 1. It correlates in the first 4 years very highly to brain volume as assessed with MRI. One would wish that the discussion would go into more detail why LAZ may be related to IQ at all and why it may be different in the different samples, i.e. in well- nourished versus a sample that may indicate malnourishment and stark drop-off in the first year. Reference 1. Jaekel J, Sorg C, Baeuml J, et al. Head Growth and Intelligence from Birth to Adulthood in Very Preterm and Term Born Individuals. Journal of the International Neuropsychological Society 2018:1-9. doi: 10.1017/S135561771800084X [published Online First: 2018/11/14]
--	--

VERSION 1 – AUTHOR RESPONSE

Reviewer 1		
1. 1	Conceptually and from the perspective of control variables and policy, it is surprise that the emphasis is completely on length/height whereas weight should do the same. In addition, it is easier to have a policy impact on weight rather than on height. Including it in the set of variables could reveal which of the variables is explaining IQs better. At least, it should be explained why weight is ignored altogether.	Length/height is the primary focus of this paper as many of the studies we describe in the introduction and discussion, including three Lancet series on child development (which are the first three references we cite in the first sentence), identify slow early childhood linear growth as a key risk factor for impaired cognitive achievement. We have altered the text to make this clear from the very start: Page 4, lines 45-47: “Three Lancet series on child development¹⁻³ have identified constrained physical growth in early-life, from conception to age 2 years, as one of the key risk factors for impaired cognitive achievement in low- and middle-income countries (LMICs)” Many of the longitudinal studies that examine this association have produced inconsistent results; and so, the goal of this paper was

		primarily methodological in demonstrating that, in fact, the inconsistency stems from the use of different statistical models for growth, and that they are in fact not discordant if we understand the contrast that each model generates. Making substantive inferences related to policy or potential interventions was only secondary. But we want to be clear that we do not believe that growth in length/height itself has a causal effect on IQ, but rather, we use it in our analysis as a proxy for undernutrition. To make this clearer, we added the following in the introduction: Page 5, Lines 77-87: “It is important to acknowledge that although we refer to these estimates as effects, where we borrow language from causal mediation, they should be interpreted as associations. It is unlikely that the relationship between linear growth and brain development is causal as we are not aware of any convincing biological mechanism for this effect. Instead, it is more likely that linear growth and brain development share the same underlying determinants, such as maternal and infant undernutrition, which is common in these settings¹⁻³. Through this lens, length/height can be viewed as a proxy (albeit imperfect) for exposure to undernutrition, where an association between early-life linear growth and cognitive scores would be indicative of a nutritional intervention that prevents both slow growth and faltering in brain development (i.e., it would represent the effect of intervening nutritionally, as opposed to the effect of physical growth itself)¹⁰.”
--	--	---

		Further, we do not preclude the assessment with other anthropometric measures; in fact, we encourage this, but our focus for this paper was primarily methodological. We added a line in the discussion to highlight that our paper is merely an illustrative example (framed by the Lancet child development series and the historical significance of using length/height growth) and that other investigators should use these methods to explore the relations with other anthropometric measures, such as weight or height, that may be more relevant in terms of policy: Page 15, Lines 314-318: “Although we use growth in length/height and IQ for our illustrative example, this approach can be extended to investigating critical windows of growth using other anthropometric measures, such as weight or head circumference (which may be more relevant for cognitive achievement^{46,47}) and other later-life health outcomes.”
1. 2	p.9: If the relation is only approximately linear, it could be useful to learn how results change if one or both of the variables had the form of natural logarithms.	Yes, this is true that different functional forms could be explored with regards to the exposure, outcome, or the association between the two. However, these are not procedures that are typically done in formulating these growth models in studies examining growth and later-life outcomes. Our goal in this paper was to try make coherent the inferences from growth models, as they are commonly formulated in the literature, to help investigators interpret results that are derived from these different growth models. Furthermore, whether the exposure or the outcome are log-transformed, they would still identify the same critical window, but we feel the

		results are much more accessible if they are presented as they currently, as the change in the outcome associated with a one-unit change in LAZ is much more digestible than one with a one-unit change in $\log(\text{LAZ})$.
1.3	If I understand this correctly, IQs are measured only once in each of the samples, and therefore this is a cross section analysis. The text about longitudinal analysis, in contrast seems to indicate that there is a time dimension, leading to a panel analysis. This could be pointed out more clearly.	Yes, IQ was measured only once in mid-childhood, but this is still not a cross-sectional analysis, as there is serial anthropometry data (that is, we use multiple measurements of length from birth to mid-childhood). We describe this throughout the text, and we already have a section outlining the number and timing of each of these measures “Linear growth exposures” in the Methods section of the paper (pages 6-7, lines 106-117).
1.4	Stata can provide VIF (variance inflation factors) which signal collinearity with control variables. In case of collinearity, the coefficients may change and this may disturb the view on ‘sensitive windows’. The standard recipe than is to drop the weaker variables. In connection with the issue of weight, indicated above, this could provide much more clarity. If close to windows, the use of polynomial distributed lags may bring the shape forward. If this would be similar across the three samples, it would really be an interesting result.	The variables we chose as potential covariates were specified a priori as potential factors that could confound the relationship between growth and IQ. It is standard practice in epidemiology to not drop variables based on statistical criteria, as we are concerned with the magnitude of the estimates rather than the precision of these estimates. For example, education and employment are likely highly correlated, but both should be in the model to control for socioeconomic status, as only having one of them may not fully capture this underlying construct where part of the growth-association may be explained by socioeconomic status. Thus, we recognize that there is a bias-variance trade-off, but we are looking for the most internally valid estimate rather than the most parsimonious model with the smallest variance. We also recognize that a polynomial distributed lag model is another appealing option to model direct effects, but again, the goal of this

		paper was to explain the inconsistencies based on the three most commonly used models found in the literature. However, we appreciate the comment and added a line to make clear that there may be other methods to estimate these effects: Page 15, lines 301-303: “We recognize that this approach has its limitations, and that there are many alternative methods to model growth⁶; for example, a distributed lag model may provide insights into sensitive windows by modeling direct effects as a smooth function of age⁴⁷.”
1.5	Most importantly, Table 3 should have or t (z) values, and indicator to let us see something of the goodness of fit. In sum, there is much room of improvement here. In economics, it would be standard to use this before publication. In connection with differences in protocol procedure the editor should decide what the authors have to do and what is a suggestion for future research.	Yes, we recognize that presenting test-statistics, such as the t- or z-statistic, and their corresponding p-values is common practice in economics, but it is not the norm in epidemiology/medicine. The standard practice is to present 95% confidence intervals to indicate the range of estimates from these models that are compatible with the data. Thus, we do not feel that it is appropriate to present the test-statistics, as the 95% confidence intervals give at least the same information (as it is just an inversion of the test-statistic and so would be redundant) and also, it would go against common practice in epidemiology/medicine.
1.6	Define ‘stature’	We used “stature” and “size” interchangeably in the paper, where stature was not defined. Thank you for pointing this out, as it can be confusing. So, we have decided to just use one of these terms, and therefore, all instances where we use “stature”, they have been replaced with “size”.
1.7	It remains unclear why the initial value is interpreted as revealing information on postnatal growth. That comes after the	If we understand this comment correctly, this refers

	initial values and would have to be excluded as an extra variable as it is done in the second and third approach. If I misunderstand this, it would be helpful to get to know what medical sciences think about the causal role of interval 1-4 for LAZ.	to the interpretation of the initial value for the change score model which can be interpreted as the effect of growth in the prenatal and postnatal periods. The reason why this is interpreted as such is because of how the model is parameterized. By including changes in postnatal size in the model, we have to interpret the coefficient for initial size, as if we are holding these changes in postnatal size constant. That is, we are constraining all the variability to be contained within initial size, such that it would represent the association for both prenatal and postnatal growth (as we are artificially setting postnatal growth to be zero in the parameterization). We agree this is a strange model, and we talk about how it is non-intuitive, and how interpreting the coefficients warrants caution, especially given that its use in the literature is relatively frequent: Pages 12-13, lines 240-251: “Investigators who elect to use the change score model alone should be careful to correctly interpret its coefficients as estimates of the cumulative direct effect of growth for a given period and all successive periods included in the model. This is a non-intuitive interpretation, and it is important to recognize that the model coefficients cannot be interpreted as estimates of the discrete effects of growth in the interval from which they were derived. As demonstrated here, the change score model coefficients for birth size represented the theoretical maximum direct effect of a one-unit increase in LAZ on IQz in all three cohorts; concordant with expectations, the coefficients then decrease
--	---	---

		in magnitude as the period of growth approaches the time of outcome assessment. Investigators may be tempted to attribute the strongest effects of growth to the earliest ages; however, coefficients from the other two models revealed that the direct and total effects are largest from mid-infancy to late infancy in CPP, and late infancy to mid-childhood in both PROBIT and CLHNS.”
Reviewer 2		
2.1	I enjoyed reading your paper and found it exceedingly well written with a balanced view of the subject. You have adequately discussed the possible drawbacks of the study. I have very little to add to this. However, a specialist review for the statistical analysis of the paper would be advised.	Thank you!
Reviewer 3		
3.1	This study sets out to compare the relationship of the method of computing growth using three different models and the relationship to IQ in mid childhood. In a nutshell, all three models: life course, conditional change and change score came in two cohorts to more or less similar findings; i.e. they mostly indicated in the three cohorts studied similarly the time periods when growth had the most effect. Quite different findings were found between the Lifecourse model and the other two models in their association to IQ. In the CLHNS the findings were quite different. One could argue that it would not matter what data are put in as long as the three models use the same data. But the authors also compare findings across the three cohorts that were sampled across different areas. And the critical windows found differ for these three cohorts. The question is how can that be explained?	Yes exactly, this was primarily a methods paper demonstrating the integration of the three most commonly used growth models, as they are formulated in the literature, to provide coherent inferences on the growth-IQ association, so we agree that the cohorts themselves were not important for the methodological demonstration.
3.2	To have a fair comparison, the three cohorts should be harmonised. Although differently measured this was done via z-scores for growth (LAZ) and IQ outcome via WHO standards or within cohort, respectively. However, different confounders were included and those which measured a similar construct were not harmonised to measure the same. If they cannot be harmonised they should be discarded to not compare "apples with bananas".	We agree that the adjustment of different covariates across the three cohorts is a limitation of the harmonization process. However, most of the confounding for these associations occur through socioeconomic status, so behavioral factors such as maternal smoking would be adequately controlled for by controlling for socioeconomic status. Therefore, we do not always need to control for the same variables in order to get internally valid estimates (see comment below for more on confounding by maternal smoking/behavioral traits and revisions to the text).
3.3	There are major confounders that may explain differences – such as the effect of prenatal growth in the oldest sample (USA). A major confounder related to growth is maternal	We agree that maternal smoking may be a confounder as it is associated

smoking or passive smoking exposure. The rate of smoking was the highest in the USA in the 1950s and 60s in documented history (https://www.cdc.gov/mmwr/preview/mmwrhtml/mm4843a2.htm). Thus foetuses may have been strongly compromised. Thus smoking in all three cohorts as well as breastfeeding rates (as related to growth) should have been included as confounders or critically discussed why they were not included.	with growth and mid-childhood IQ. However, maternal smoking is only associated with IQ through its association with socioeconomic status, which has been shown in prior research (not only in these cohorts themselves, but also in other settings as well). Thus, we agree that fetuses may have been strongly compromised due to smoking, but this would occur non-differentially with respect to the outcome conditional on socioeconomic status (that is, it will not bias the estimates if socioeconomic status was controlled for). Since we agree that this was initially unclear, and is an important discussion point, we have added the following to the text: Page 14, lines 280-295: “Although we controlled for an extensive, albeit different across the three cohorts, panel of baseline covariates (i.e., child, parental and household attributes), we may not have controlled for all relevant confounders. For example, we did not control for maternal smoking³⁷⁻⁴⁰, which was highly prevalent in the United States in the 1950s-60s⁴¹. However, maternal smoking is only associated with mid-childhood cognitive achievement through its association with socioeconomic status, as shown by previous research in the same cohorts examined in the present study^{42,43}, and in other settings^{44,45}. One possible explanation of this finding is that maternal smoking affects early-life brain development^{39,40}, but the effects diminish through childhood, such that by mid-childhood (i.e., timing of outcome assessment) smoking is just a behavioral trait embedded in a broader constellation of social factors
---	--

		that influence child brain development, such as early stimulation and learning opportunities ³ . Thus, our control for parental education, employment and socioeconomic status quintile in our models would be sufficient to block this backdoor path (borrowing language from causal graphs ⁴⁶) through maternal smoking (and any factors associated with mid-childhood cognitive achievement that are largely shaped by a mother's social conditions).”
3. 4	The number of those lost to follow-up is large in all samples but this is not so concerning as this study is about comparing different predictive models. And previous research has found relatively unbiased prediction despite large loss in follow-up. Nevertheless, in STATA missing data for some of the variables could have been computed with the MICE function.	We agree that multiple imputation could be a potentially useful sensitivity analysis to assess the robustness of the associations due to those lost to follow-up (as it would be for any analysis), but multiple imputation comes with many additional assumptions, such as the missingness mechanism or assumptions of the MICE algorithm (e.g., the iterative sampling from conditional distributions, whether the conditional distribution is correctly specified etc.). The way these models are commonly formulated and used in the literature are typically in a complete-case analysis, and therefore, we did not feel the need to introduce this additional layer of complexity, especially given that these methods are likely to give unbiased results, as you have mentioned.
3. 5	The most concerning is there should be some explanation why LAZ should be related to IQ and via what mechanism. IQ is a measure of brain function and how should weight and height relate to it – which biological or social mechanisms? Intelligence happens in the brain and thus, of the different anthropometric measures, head circumference is the closest measure related to brain growth 1. It correlates in the first 4 years very highly to brain volume as assessed with MRI. One would wish that the discussion would go into more detail why LAZ may be related to IQ at all and why it may be different in the different samples, i.e. in well- nourished versus a sample that may indicate malnourishment and stark drop-off in the first year.	See response to comment #1.1 But we would also like to thank you for the reference provided, and have cited it in our revised version: Page 15, Lines 312-315: “Although we use growth in length/height and IQ for our illustrative example, this

	Reference 1. Jaekel J, Sorg C, Baeuml J, et al. Head Growth and Intelligence from Birth to Adulthood in Very Preterm and Term Born Individuals. Journal of the International Neuropsychological Society 2018:1-9. doi: 10.1017/S135561771800084X [published Online First: 2018/11/14]	approach can be extended to investigating critical windows of growth using other anthropometric measures, such as weight or head circumference (which may be more relevant for cognitive achievement^{46,47}) and other later-life health outcomes.”
--	---	---

VERSION 2 – REVIEW

REVIEWER	Thomas H.W.Ziesemer U Maastricht, Netherlands
REVIEW RETURNED	29-May-2020

GENERAL COMMENTS	Thanks for your very clear and convincing answers. Just in case you want to think about collinearity in future research, it is not only about SD but also about sign and size of coefficients.
--

REVIEWER	Dieter Wolke University of Warwick
REVIEW RETURNED	05-Jun-2020

GENERAL COMMENTS	The authors have addressed the concerns raised in the initial review in this revision and added relevant literature.
--